# Family-, School-, and Neighborhood-Level Predictors of Resilience for Adolescents with a History of Maltreatment

**DOI:** 10.3390/children10010001

**Published:** 2022-12-20

**Authors:** Yujeong Chang, Susan Yoon, Kathryn Maguire-Jack, Jihye Lee

**Affiliations:** 1School of Social Work, University of Michigan, Ann Arbor, MI 48109, USA; 2College of Social Work, The Ohio State University, Columbus, OH 43210, USA; 3Department of Social Welfare, College of Social Sciences, Ewha Womans University, Seoul 06974, Republic of Korea; 4School of Social Welfare, Chung-Ang University, Seoul 06974, Republic of Korea

**Keywords:** child maltreatment, protective factors, resilience, families, schools, neighborhoods

## Abstract

Child maltreatment is a well-known risk factor that threatens the well-being and positive development of adolescents, yet protective factors can help promote resilience amid adversity. The current study sought to identify factors at the family, school, and neighborhood levels associated with resilience outcomes including positive functioning and social skills, among adolescents who have experienced maltreatment. Using longitudinal data from the Fragile Families and Child Wellbeing Study, the analytic sample was limited to 1729 adolescents who experienced maltreatment before age 9. Family-, school-, and neighborhood-level predictors were assessed at age 9, and youth resilience was measured at age 15. We conducted a series of multiple regression analyses to examine multi-level protective factors at age 9 as predictors of positive adolescent functioning and social skills at age 15. The study found that mothers’ involvement was significantly and positively associated with positive adolescent functioning and social skills. Additionally, school connectedness and neighborhood social cohesion were significantly associated with higher levels of adolescent social skills. Our findings suggest that positive environmental contexts such as maternal involvement in parenting, school connectedness, and socially cohesive neighborhoods can serve as important protective factors that promote resilient development among adolescents who have experienced maltreatment as children.

## 1. Introduction

Child maltreatment is a global public health concern associated with severe and long-lasting detrimental outcomes across numerous domains of health, development, psychosocial functioning, and well-being throughout the child’s life, including during childhood, adolescence, and adulthood [1,2,3,4]. Numerous research studies have been published on the adverse impacts of childhood maltreatment, with well-established relations to physical (e.g., diabetes, cancer, neurological alteration), psychological (e.g., anxiety, depression, post-traumatic stress, social difficulties), and behavioral problems (e.g., criminal activities, risky sexual behavior, and substance use) [4,5,6,7]. While early studies focused on examining the etiology and devastating consequences of child maltreatment, the paradigm has shifted in recent years to understanding the potential protective factors and positive constructs that can affect outcomes after child maltreatment [8,9]. Although there is ample evidence that maltreatment has detrimental impacts on children, not all victims of child maltreatment follow maladaptive developmental trajectories [10]. Some children fare better in the aftermath of maltreatment, showing positive adaptation despite the adverse experiences they have undergone [11]. While individual characteristics (e.g., positive temperament, intelligence) can be an essential buffer to support resilient functioning [12], the environment (e.g., family, school, neighborhood) that children interact with daily is also a highly influential factor capable of empowering resilience in youths who have experienced maltreatment [13,14]. The current study sought to examine environmental (i.e., family-, school-, and neighborhood-level) protective factors as predictors of resilience in youths who have experienced childhood maltreatment.

## 2. Background

### 2.1. Childhood Maltreatment

Child maltreatment continues to be a prevalent and severe social concern, and can involve physical abuse, psychological or emotional abuse, sexual abuse, and neglect. Child maltreatment is defined by the Child Abuse Prevention and Treatment Act (CAPTA) as “any recent act or failure to act on the part of a parent or caregiver that results in death, serious physical or emotional harm, sexual abuse, or exploitation, or an act or failure to act that presents an imminent risk of serious harm” [15].

In 2020, there were approximately 3.9 million referrals to Child Protection Service (CPS) agencies for investigation regarding alleged cases of child maltreatment [16]. An estimated 618,000 children were identified as victims of child abuse or neglect [16]. Among these substantiated child maltreatment cases, the majority (76.1%) of children were neglected, a sizeable number of children (16.5%) were physically abused, 9.4% were sexually abused, and 0.2% were sex trafficked [16]. As a result of child maltreatment, nationally, an estimated 1750 children died in 2020, at a rate of 2.38 per 100,000 children [16].

Considerable research has revealed that maltreatment experiences in early life increase the risk of maladaptive consequences, with sequelae spanning a wide range of developmental domains in children and adolescents. With regard to the neurobiological realm, empirical evidence indicates that exposure to childhood maltreatment may have significant and lifelong impacts on children’s brains [17,18]. For example, a study by McCrory and colleagues (2011) found that childhood maltreatment may result in a reduced volume of the hippocampus, a central part of the brain that plays a role in memory and learning capacity [19]. Prior studies also found a decreased volume in the corpus callosum, which is responsible for higher cognitive abilities [20]; decreased volume of the cerebellum, involved in the coordination of executive functioning and motor behavior [19]; a smaller volume of the prefrontal cortex, critical to social and emotional regulation [21]; and overactivity in the amygdala, central to determining threat stimulus and triggering emotional responses [22]. Furthermore, individuals who have experienced childhood maltreatment are at higher risk of psychopathological symptoms and maladaptive socio-emotional development, leading to relationship challenges and behavioral problems. For example, early maltreatment experiences are associated with mental health problems including anxiety and depression [6,7,23,24], relationship problems including peer and romantic relationships [25,26], antisocial behavior leading to delinquency and crime [27,28,29], and problems relating to substance use [30,31].

### 2.2. Child Maltreatment and Resilience

While substantial research has demonstrated various damaging developmental outcomes of childhood maltreatment, some children and adolescents adapt to early life adversities and follow a positive and healthy developmental pathway despite past maltreatment experiences. The ability of these children and adolescents to adapt to adverse early life experiences prompted researchers to shift their focus from studying the risk factors and adverse outcomes of child maltreatment to understanding the protective contexts and positive constructs that help to mitigate the impact of child maltreatment.

While studies on resilience among individuals with histories of childhood maltreatment have evolved over the years, there has been little consistency in how resilience is conceptualized and operationalized [2,9]. Early studies conceptualized resilience as a somewhat inherent and stable individual-level personality trait characterized by ego resilience and the ability to cope in stressful situations [32]. On the other hand, other studies have conceptualized resilience as a two-dimensional construct that represents positive adaptation (e.g., social competence, achieving stage-salient developmental tasks) following adverse experiences [33]. Additionally, various studies have conceptualized resilience as the absence or lower levels of adverse behavioral outcomes by measuring different indicators of psychopathology such as externalizing symptoms [34,35], antisocial behavior [36], and aggression [37]. In such studies, individuals who scored in the normal or lower range on measures of psychopathology were regarded as resilient.

Ungar (2004) described an ecological approach to resilience, where resilience is conceptualized as individuals despite adversities demonstrating qualities associated with a normative definition of health, compared with a constructionist approach that provides an innovative framework conceptualizing resilience [38]. In the constructionist approach, resilience is conceptualized as a successful outcome through negotiation between an individual and the environment, that provides resources to define oneself as healthy despite adversity [38]. In other words, Ungar (2004) conceptualizes resilience within the discourse of people navigating their ways to external resources that strengthen their well-being despite adversity [38]. In a more recent study, Ungar (2015) suggests a multidimensional assessment of resilience by incorporating individual qualities (e.g., personality, cognition, temperament), contextual dimensions of adaptive functioning, and cultural factors [39].

In recent studies of resilience following childhood maltreatment, multiple domains are involved in operationalizing the construct of resilience [9]. For example, in a study by Dubowitz and colleagues (2016), resilience was operationalized through three broad domains: behavior, social, and cognitive development [40].

Generally, resilience is understood as a dynamic developmental process involving the attainment of positive adaptation despite experiences of severe adversity or trauma [33]. Although the literature on “resilience” in the context of child maltreatment differs in conceptualizing and operationalizing resilience processes [9], studies indicate that profiles of resilience in children with maltreatment histories are affected both by traits of the surrounding environment as well as individual attributes [41].

As noted above, in the literature to date, there is a lack of consensus on the conceptualization of resilience. While some literature conceptualizes resilience as a personality trait, some as lower levels of adverse outcomes, and some as the accessibility and ability to acquire socioecological resources, numerous studies conceptualize resilience as adaptive functioning in the face of adversity [9]. For the purpose of our study focusing on resilience in adolescents who have experienced childhood maltreatment, we sought to apply a strengths-based perspective for conceptualizing resilience. Adolescence is a critical developmental stage when adolescents begin to build various relationships outside their family setting and expand their social networks, and also a time when they spend much of their time in the school setting. As a result, for our conceptualization of resilience, we focused on functioning and skills relevant to the school setting, including positive psychological functioning (e.g., engagement, perseverance, optimism, connectedness, and happiness) and social competence (e.g., adaptive social skills).

### 2.3. Multi-Level Predictors of Positive Adolescent Functioning and Social Skills

From the perspective of developmental psychopathology, the current study is guided by bioecological theory and the ecological–transactional model. Bronfenbrenner’s bioecological theory of human development (1979) demonstrates the complexity of understanding various outcomes in children who have experienced maltreatment [13]. The bioecological theory emphasizes the interrelated dimensions of social ecology that locate a child within their surrounding environment, indicating that multiple-level factors contribute to children’s outcomes [13,42]. Similarly, from the perspective of developmental psychopathology, the ecological–transactional model by Cicchetti and Lynch (1993) acknowledges that individuals are embedded in their surrounding environments and that individuals interact with and respond to factors at each ecological level (i.e., family, community, culture, and the transactions between those levels), allowing for differences in developmental outcomes [14,43]. The developmental pathways for children experiencing maltreatment to either maladaptive or adaptive functioning are affected by the intricate dynamics of individual-, familial-, community-, and broader macrosystem-level risks and protective factors in the wake of those early experiences [44]. Drawing from the aforementioned theories, various individual and contextual (e.g., relational-level, community-level) protective factors may predict positive developmental functioning in children with histories of maltreatment.

At the individual level, a study by Cicchetti et al. (1993) showed that ego overcontrol, ego resilience, and positive self-esteem served as protective factors that promote competent functioning in children who had undergone early experiences of maltreatment [45]. Simply stated, ego overcontrol is a personality trait representing how well a person manages aggressive impulses, particularly under stressful situations, and ego resilience refers to how well an individual adapts to and recovers from adverse situations while preserving their identity [46]. High cognitive abilities (e.g., higher IQ) were also identified as a protective individual-level factor predicting resilience in children following child maltreatment [47,48,49]. For instance, a study by Herrenkohl and colleagues (1994) based on longitudinal data, reported that among 23 individuals who were categorized as the “resilient” group showing higher levels of functioning following childhood maltreatment, 14 individuals were of average or above-average IQ [47].

At the relationship level, the theory of attachment presented by Bowlby (1969) set a theoretical foundation by demonstrating that a close and positive relationship with caregivers helps to form a secure base for children’s development of self-worth, trust, and socially adaptive skills in forming and maintaining relationships with others [50,51]. Empirical studies have also found evidence for close and positive parent–child relationships [4,52], maternal warmth [53], and parental well-being (i.e., absence of problems with mental health and substance use) [37] as protective factors related to resilience following childhood maltreatment. For example, from a systemic review by Meng and colleagues (2018) examining resilience and protective factors in individuals with histories of childhood maltreatment, it was found that maternal care and close mother–child relationships served as consistent protection in these individuals, allowing for a better life and adaptive functioning [52]. Moreover, studies indicate that positive parenting styles and parent–child relationships, in general, foster the formation of adolescents’ social skills [54]. Collectively, theoretical and empirical evidence shows that supportive, caring, and stable parent–child relationships and family environments can reduce the adverse impacts of child maltreatment and help foster adaptive functioning in children following child maltreatment [1].

Because children and adolescents spend considerable time in school and in their surrounding neighborhoods, it is crucial to understand how school and neighborhood–level factors can serve as protective factors at the community level among youths with early adverse experiences. In this regard, studies have found protective factors related to school [34,55] and neighborhood [36,56] that predict positive functioning among children with early experiences of maltreatment. For instance, a study by Resnick (1997) demonstrated that a higher level of family connectedness and school connectedness predicted positive adaptive functioning in a national longitudinal sample of adolescents encompassing multiple behavioral domains (i.e., emotional distress, suicidality, violence, sexual behaviors, and substance use) [57]. Similarly, another longitudinal study found that a strong commitment to school predicted less delinquency, lower violence, and fewer status offenses in adolescents regardless of childhood maltreatment history [58].

Neighborhood factors include neighborhood safety, a sense of connectedness and belonging, socioeconomic status, and collective efficacy. A comprehensive review of the literature on the influences of neighborhood factors on children and adolescents’ well-being reported that neighborhood socioeconomic status was positively related to various indicators of functioning in adolescents [59]. Additionally, a study by Jaffee and colleagues (2007) found that children living in neighborhoods with lower crime rates and higher levels of collective efficacy showed lower levels of antisocial behavior and more resilience following childhood maltreatment [36].

### 2.4. The Current Study

While ample research has examined resilience development in children and adolescents who experienced maltreatment in early life, comparatively less consideration has been given to the different contexts in which children could be supported. Filling this gap is particularly critical to develop, determine, and implement the best interventions to ensure optimal development and promote resilient functioning across multiple developmental domains in youths who have experienced child maltreatment. Furthermore, such a study would help prevent adverse developmental cascades in children with histories of maltreatment. The current study aimed to fill existing gaps in the literature and contribute to supporting children and adolescents with maltreatment histories. In this context, we examined multi-level factors (i.e., family, school, and neighborhood levels) in a longitudinal sample of children who had been maltreated at or before they were 9 years old, drawn from the Fragile Families and Child Wellbeing Study (FFCWS). Specifically, our research aim was to examine protective environmental factors encompassing family (i.e., maternal involvement and paternal involvement), school (i.e., school connectedness), and neighborhood- (i.e., social cohesion and informal social control) contexts that are associated with positive adolescent functioning and social skills, among adolescents who experienced maltreatment at or before age 9. We aimed to examine three research hypotheses. First, given previous literature suggesting that positive parent–child relationships [4,52] and maternal care [53] serve as protective factors associated with resilience for individuals who have experienced childhood maltreatment, we hypothesized that parents’ involvement in parenting would be positively associated with resilience in adolescents with a history of childhood maltreatment. Second, based on previous research [34,55,58], we anticipated that higher school connectedness would be associated with resilience. Third, we hypothesized that higher levels of social cohesion and informal social control would be associated with resilience, aligning with previous literature on neighborhood-level factors affecting resilience following childhood maltreatment [36,56].

## 3. Methods

### 3.1. Sample and Procedures

The current study used data from the Fragile Families and Child Wellbeing Study (FFCWS). FFCWS is a longitudinal birth-cohort study of approximately 5000 children born in large urban cities in the United States between 1998 and 2000, followed through early adulthood [60,61]. The initial goal of FFCWS was to explore the characteristics and relationships of unmarried parents, how policies impacted these relationship dynamics, and how parents and children fared in these families [61]. As new types of data and technology were incorporated into the study, its topics have been broadened to conceptualize family wellbeing [60]. FFCWS used an intricate, multistage sampling procedure for recruitment, including participants from diverse family types, racial and ethnic groups, socioeconomic statuses, social contexts, etc. [60]. The sampling procedure resulted in an oversampling of non-marital births, as births to unmarried couples tend to occur disproportionately among disadvantaged groups [60]. Accordingly, the sample comprised many low-income and minority families, including Black and Hispanic families [60,61].

Data were collected following the birth of the focal child (baseline; wave 1) and followed up when the focal child was aged 1 (wave 2), 3 (wave 3), 5 (wave 4), 9 (wave 5), 15 (wave 6), and 22 (wave 7; in progress) [60]. In addition to using medical records, the collection of baseline data was conducted in a survey format with mothers and fathers in hospitals shortly after the focal child was born (wave 1) [60]. Mothers and fathers of the focal child were surveyed again when the focal children were aged 1, 3, 5, and 9 [60]. During waves 3 and 4, FFCWS investigators conducted home visits, including interviewer observations, assessments, and a survey of primary caregivers [60]. The staff of FFCWS observed childcare settings and surveyed primary caregivers for children who did not have maternal kin or who received center-based care [60]. Home visits were also held during wave 5, but a brief child survey was carried out instead of a primary caregiver survey [60]. A teacher survey was also conducted for children in kindergarten during wave 4 and all children in wave 5. During wave 6 (adolescence), adolescents and primary caregivers were surveyed, and approximately 1000 families were randomly selected for home visits that included interviewer observations and assessments [60]. Wave 7 is currently in progress, including surveys of the focal children, now young adults, and surveys of the primary caregivers who cared for the focal child during wave 6 [60].

The analytic sample of the present study was limited to samples in which the focal child was maltreated at or before age 9, including waves 3, 4, and 5 when the focal child ages were 3, 5, and 9. We limited the sample to children who had any maltreatment reported by the child at age 9, for a total sample of 1729.

Maltreatment was measured using the Conflict Tactics Scale—Parent to Child version (CTS-PC) [62]. When children were aged 3, 5, and 9, primary caregivers were asked a series of 15 questions related to physical abuse, psychological abuse, and neglect, relating to behaviors in the past year. We restricted the sample to children whose primary caregiver reported using any abusive or neglectful behavior in the preceding year, when the child was aged 9 or younger.

### 3.2. Measures

#### 3.2.1. Positive Adolescent Functioning

Positive adolescent functioning was assessed at wave 6, when the focal child was aged 15, using the adapted version of the EPOCH Measure of Adolescent Wellbeing scale [63]. The adapted version of EPOCH assessed five items: engagement, perseverance, optimism, connectedness, and happiness. Four questions were asked in order to assess each of the five items; example items include: “There are people in my life who really care about me” to measure connectedness and “Once I plan to get something done, I stick to it” to measure perseverance. Adolescents were asked to rate how much they agreed with each of the five items referring to the past four weeks, using a 4-point set of response options (1 = Strongly agree to 4 = Strongly disagree). Items were recoded so that higher scores indicated a higher level of functioning (1 = Strongly disagree and 4 = Strongly agree). The mean was calculated to create a total score. The internal reliability of this measure in our sample (Cronbach’s alpha) was acceptable (α = 0.80).

#### 3.2.2. Adolescent Social Skills

Social skills of the adolescents were assessed at wave 6, when the focal child was aged 15, using an adapted version of the Express subscale of the Adaptive Social Behavior Inventory (ASBI) [64] and the Assertion scale of the secondary-level parent and teacher forms of the Social Skills Rating System (SSRS) [65]. Three items were adapted from the ASBI, for example: “I am open and direct about what I want”. The ASBI was originally designed for educators to report on children’s social skills. The questions were modified to be appropriate for adolescents’ self-reporting [64]. Nine items were adapted from the SSRS. Example items include: “I join group activities without being told to”, “I start conversations rather than waiting for others to talk first”, and “I report accidents to appropriate persons”. The SSRS originally asks the frequency of children’s behavior, and how important the respondent considers certain behaviors that are critical to children’s development, each on a 3-point scale (0 = never, 1 = very often or not important, and 2 = critical) [65]. These items were modified, excluding the question about importance, and asking adolescents to rate the truth of each statement for them, instead of frequency of behavior, on a 3-point scale (1 = not true, 2 = sometimes true, and 3 = often true). Item responses were recoded so that 0 = not true, 1 = sometimes true, and 2 = often true, and were respectively summed to create a total score for each. The internal reliability in this sample was acceptable (α = 0.74).

#### 3.2.3. Mother’s and Father’s Involvement

Mothers’ and fathers’ respective involvement in parenting were assessed using a six-item scale at wave 5, when the focal child was age 9. Example items include: “Talk over important decisions with you” and “Spend enough time with you”. Adolescents selected from a 4-point response scale the extent to which each statement described their impression of their parents’ involvement in parenting. Item responses were recoded so that higher scores indicated a greater degree of involvement in parenting. Items were summed to create a total score. The internal reliability of this measure by parent type was somewhat low (father: α = 0.75, mother: α = 0.55).

#### 3.2.4. School Connectedness

School connectedness was assessed at wave 5, when the focal child was age 9, using adolescents’ self-reporting. Four items were compiled by Jacquelyn Eccles for the Panel Study on Income Dynamics–Child Development Supplement (PSID-CDS-III) [66] to assess the degree of inclusiveness, happiness, closeness, and safety that adolescents experience at school; for example: “How often did you feel like you were part of your school?” Each item was rated on a 5-point Likert scale ranging from 0 = not once in the past month to 4 = every day. Items were summed to create a total score. These four-item measures had adequate internal reliability (α = 0.70) with scores ranging from 0 to 4.

#### 3.2.5. Social Cohesion and Informal Social Control

According to Sampson and colleagues [67], neighborhood social cohesion and informal social control are the two core components of collective efficacy. Neighborhood social cohesion refers to the mutual trust, connectedness, and solidarity between residents in a neighborhood [68]. On the other hand, informal social control refers to the willingness of residents in a neighborhood to intervene in situations that they suspect would be problematic when occurring within their neighborhood [69].

The current study assessed neighborhood social cohesion and informal social control as neighborhood-level factors. Research by Barnhart and colleagues [70] suggested that understanding social cohesion and informal social control as two separate constructs of collective efficacy can allow better estimation of the benefits brought by each construct.

Neighborhood social cohesion was measured at wave 5, when the focal child was aged 9. The survey of primary caregivers included four items that assessed levels of cohesion and trust within the neighborhood, for example: “People around here are willing to help their neighbors.” The items were adapted and modified using measures developed by Sampson and colleagues in the *Project on Human Development in Chicago Neighborhoods (PHDCN): Community Involvement and Collective Efficacy* [67]. Each item was rated on a 4-point Likert scale ranging from 1 = strongly agree to 4 = strongly disagree. Item responses were recoded so that higher numbers indicated higher levels of social cohesion, and items were summed to create a total score. These four-item measures had adequate reliability (α = 0.79) with scores ranging from 1 to 4.

Neighborhood informal social control was assessed at wave 5, when the focal child was age 9. The survey of primary caregivers included five items measuring informal social control within the neighborhood, for example: “If children were skipping school and hanging out on the street.” Informal social control items were also adapted and modified according to the same measures of neighborhood social cohesion. Each item was rated on a 4-point Likert scale ranging from 1 = very likely to 4 = very unlikely. Item responses were recoded that higher numbers indicated higher levels of informal social control, and items were summed to create a total score. These five-item measures had adequate reliability (α = 0.87) with scores ranging from 1 to 4.

#### 3.2.6. Control Variables

The current study controlled for a variety of covariates, including child’s gender, child’s race or ethnicity, mother’s age, mother’s education, mother’s marital status, number of children in the household, and economic hardship. Child’s gender (*0 = female, 1 = male*) and race/ethnicity were reported at baseline. For child’s race or ethnicity, White was used as the reference group, and Black, Hispanic, multiracial, and other were included as dummy variables. Maternal age was assessed in numeric values at baseline. Maternal education was assessed based on self-reporting when the focal child’s age was 9. These items were recoded as dichotomous variables (0 = high school education or higher, 1 = less than high school). Mother’s marital status was assessed at focal child’s age 9, based a combination of variables. If at age 9 the child lived with the mother or father half or more of the time and the mother or father was married to the other parent or another partner, we considered the child to be living with two married adults. These data were recoded as a dichotomous variable (0 = not married, 1 = married). The total number of children when the focal child’s age was 9 was measured in numeric values reported by the primary caregiver. Lastly, economic hardship at focal child’s age 9 was assessed according to 10 possible items from the mother’s self-report. We summed these items to create a countable variable.

### 3.3. Data Analysis

Prior to conducting primary analyses, we first examined frequency distributions and univariate descriptive statistics, including mean, standard deviation, skewness, kurtosis, and range, to identify outliers and invalid values within the data. We also calculated bivariate correlations between each variable to examine bivariate relationships between predictors and the outcomes and to check potential problems of multicollinearity. Correlation above 0.80 was considered an indicator of multicollinearity [71]. We conducted multiple regression to identify multi-level (i.e., family-, school-, and neighborhood-level) predictors associated with positive outcomes among young people with a history of maltreatment in childhood. For multiple regression analyses, the outcome variables were positive adolescent functioning and adolescent social skills, and the focal predictors were involvement of mother or father, school connectedness, neighborhood social cohesion, and informal social control. The outcome variables were regressed on the set of predictors and control variables. Separate regression models were estimated for each outcome variable. Data analysis was conducted using STATA v.17.

## 4. Results

### 4.1. Descriptive Statistics

Table 1 presents sample characteristics of our analytic sample who were maltreated by age 9, and descriptive statistics of our key study variables. Slightly more than half of the children in our sample were male (51.5%). Regarding race and ethnicity, 46.4% of the children were Black, proportions that were White and Hispanic were each respectively 18.7%, and 14.8% were multiracial. The remaining 1.3% identified their race or ethnicity as “other”. The majority of the mothers in the sample (80.7%) had an education level above high school. The ages of the mothers in the sample ranged from 15 to 43 years (M = 25.3, SD = 6.0). About 43% of mothers were married when the focal child was aged 9. The total number of children in the household when the focal child’s age was 9 ranged from 0 to 8 (M = 2.7, SD = 1.3). Out of a total possible 10 economic hardships, families had experienced an average of 1.33 economic hardships in the past year (M = 1.3, SD = 1.7). Table 2 presents bivariate correlations between key study variables.

### 4.2. Predictors of Positive Adolescent Functioning

Table 3 presents the findings of the regression models examining multi-level protective factors (i.e., familial, school, and neighborhood levels) at age 9 as potential predictors of positive adolescent functioning and adolescent social skills at age 15. Among the key predictors, maternal involvement in parenting was the sole predictor that was significantly related to positive adolescent functioning (B = 0.86, SE = 0.32, *p* = 0.006). Black youths had higher levels of positive adolescent functioning compared to youths of other races and ethnicities (B = 1.56, SE = 0.45, *p* = 0.001), and boys had higher levels of positive adolescent functioning compared with girls (B = 1.46, SE = 0.31, *p* < 0.001).

### 4.3. Predictors of Adolescent Social Skills

Maternal involvement in parenting was significantly associated with higher levels of adolescent social skills, as were school connectedness and social cohesion (mother’s involvement: B = 0.37, SE = 0.19, *p* = 0.049; school connectedness: B = 0.31, SE = 0.10, *p* = 0.001; social cohesion: B = 0.38, SE = 0.16, *p* = 0.015). Black and Hispanic youths had lower levels of adolescent social skills compared with White, multiracial, and other races (Black: B = −0.55, SE = 0.27, *p* = 0.040; Hispanic: B = −0.75, SE = 0.31, *p* = 0.017). Maternal educational status lower than high school was associated with lower levels of adolescent social skills (B = −0.55, SE = 0.24, *p* = 0.024) and the mother’s total number of children was negatively associated with the focal child’s adolescent social skills (B = −0.20, SE = 0.07, *p* = 0.004). Economic hardship was also negatively associated with adolescent social skills (B = −0.13, SE = 0.06, *p* = 0.028).

## 5. Discussion

Despite the increasing volume of research examining resilience among adolescents who have experienced childhood maltreatment, it remains unclear which protective factors across the socioecology contribute to different aspects of resilience, including psychological well-being and positive social development. Drawing from the ecological––transactional model [14], the current study sought to fill this gap by examining the associations between environmental-level protective factors (i.e., family, school, and neighborhood levels) and resilience (i.e., positive psychological functioning and social skills) in young people with a history of childhood maltreatment. Specifically, we focused on mothers’ and fathers’ involvement in parenting, as well as school connectedness and collective efficacy (i.e., social cohesion and informal social control) as socio-ecological predictors of interest.

At the family level, mothers’ involvement was found to be a significant protective factor both for positive functioning (i.e., psychological well-being) and social skills. These findings affirm a robust body of evidence that suggests positive mother–child relationships represent a salient protective factor associated with healthy development in adolescents after exposure to maltreatment [1,68]. Numerous empirical studies and relationship-based theories, such as attachment theory [50,72], have emphasized the protective effects of secure mother–child attachment and mothers’ warm, reliable, and responsive care on resilience against the development of psychopathology in contexts of maltreatment [53,73,74]. Our findings expand on prior works by demonstrating that positive maternal involvement is not only associated with reduced risk of adolescent psychopathology but is also linked to positive developmental outcomes including psychological well-being and prosocial skills. Positive maternal involvement may help adolescents to develop positive representational models of the self and others, leading to favorable outcomes including adaptive patterns of psychological functioning, interpersonal relations, and social adjustment [50,75]. Such information is critical for understanding the full scope of the importance of the bond between mothers and their children. The majority of the literature has been from a deficit perspective, focusing narrowly on the ways in which subpar parenting can lead to detrimental outcomes for children. The current study assessed resilience and therefore contributes to understanding the ways in which mothers can contribute to positive outcomes in children. This information may be very empowering for mothers and parental education programs.

Surprisingly, fathers’ involvement was not found to be associated with either of the resilience outcomes examined in the study. Our findings diverge from previous findings that indicate the protective effects of fathers’ positive involvement on adolescent outcomes [76,77,78]. This may be at least in part attributed to the fact that 93% of the adolescents in our study were not living with their fathers at the time of the survey. The items used to assess fathers’ involvement may not have been applicable or effective in capturing unique patterns of involvement in non-residential fathers [79]. Utilizing more nuanced measures of fathers’ involvement may be helpful to clarify the effects of fathers’ involvement on resilience among youths with a history of childhood maltreatment [80].

Because adolescents’ experience extends beyond the home environment, and various factors outside the family context influence adolescents’ functioning and outcomes, it is critical to examine extra-familial contexts such as schools and neighborhoods as sources of resilience. At the school level, we found that school connectedness was related to higher levels of social skills in adolescents who have experienced maltreatment. This finding provides further support for findings from earlier work demonstrating school connectedness as critical for promoting positive outcomes, including in those with exposure to adversity and trauma [34,55,58,81,82]. It should be noted that many prior studies have focused on understanding the protective effects of school connectedness in preventing or reducing mental health problems or behavioral problems, including depression [83], conduct problems [84], externalizing symptoms [85], and substance use [86]. Our findings further add to the existing literature by newly illustrating a link between school connectedness and social skills in adolescents who have experienced child maltreatment. These findings suggest that helping students develop and strengthen their sense of connectedness to school may be beneficial in promoting positive social functioning among young people who have experienced child maltreatment.

At the neighborhood level, neighborhood social cohesion was associated with higher levels of social skills. This finding is in line with a robust body of research suggesting that neighborhoods with greater social cohesion and strong ties among residents predicted positive youth development outcomes including lower rates of depression, anxiety, suicidal ideation, aggression, and conduct disorder, in the context of exposure to maltreatment and/or other stressful life events [87,88,89]. Drawing from the social learning perspective [90], adolescents living in socially cohesive neighborhoods may develop positive social skills by observing and modeling neighbors who share with each other mutual trust, connectedness, and social support. Interestingly, informal social control did not show the same positive effects on adolescents’ social skills. Informal social control relates to neighbors’ willingness to step in and intervene when social problems are observed. While social cohesion deals with relationships between individuals and requires interaction, social control can occur with very little face-to-face interaction. For example, it could involve calling the police when a crime is being witnessed. As a result, social control may be less related than social cohesion to the development of social skills. More research is needed to understand the different ways by which social cohesion and informal social control influence resilience in adolescents with a history of maltreatment.

In terms of control variables, Black children were more likely to report higher levels of positive adolescent functioning but lower levels of adaptive social skills. Hispanic children were also comparatively more likely to report lower levels of adaptive social skills. It is possible that these results were affected by measurement issues. Adaptive social skills measurements included questions such as “I join group activities without being told to,” “I start conversations rather than waiting for others to talk first,” and “I report accidents to appropriate persons”. It is possible that these questions are not culturally sensitive and therefore children with different racial or ethnic identities were less likely to report these behaviors. In reviewing the three items specified above, it is possible that Black and Hispanic children, who are more likely to experience racism in schools and their communities, may be less likely to engage in these behaviors for fear of reprisal. Measurements of positive adolescent functioning, on the other hand, included such questions as “There are people in my life who really care about me” and “Once I plan to get something done, I stick to it”. In terms of face validity, these items may be less suitable measurement issues due to cultural differences in experiences.

Together, our findings illustrate the important roles played by various protective factors across the social ecology in shaping resilience during adolescence. It is noteworthy that only mothers’ involvement (i.e., a family-level protective factor) was associated with adolescents’ positive psychological functioning, while mothers’ involvement, school connectedness, and social cohesion, respectively representing family-, school-, and neighborhood-level protective factors, were associated with adolescents’ prosocial skills. Our findings might suggest that intra-familial protective factors are particularly important for intra-personal resilience (i.e., positive psychological characteristics and inner well-being), whereas both intra- and extra-familial protective factors contribute to interpersonal resilience (i.e., social skills).

## 6. Limitations

There are important limitations to this study that should be considered. The study sample primarily consisted of young people born to unmarried mothers in urban situations, limiting the study findings’ generalizability to more affluent or rural youth. Several methodological limitations were also relevant. Although we sought to ensure temporal ordering of the study variables by using predictors at age 9 and outcomes at age 15, causal inferences cannot be made about the effects of family, neighborhood, and school contexts on adolescents’ resilience as this study examined only associational effects. Additionally, the internal consistency of the scale for mothers’ involvement was poor (α = 0.55), and the findings related to maternal involvement should be interpreted with caution. Finally, we did not test the interaction effects between family, neighborhood, and school contexts, although, in reality, these contexts are likely to interact with each other in shaping resilience in adolescents. Examining the interplay among multi-level protective factors was beyond the scope of the current study, and future research should consider investigating the interactions among protective factors.

## 7. Implications

Despite its limitations, the current study offers important implications for policy and practice. Given that positive maternal involvement was associated with both intra-personal resilience (i.e., positive psychological characteristics and well-being) and inter-personal resilience (i.e., social skills), intervention programs should support the development and maintenance of strong mother–child relationships for adolescents with a history of childhood maltreatment. Furthermore, the findings on adolescents’ social skills point to the need for practitioners to adopt a socioecological framework perspective and conduct a comprehensive assessment of adolescents’ environmental contexts to maximize multi-level resources and protective factors. Based on our findings, interventions that help young people build positive relationships with their parents (i.e., family-level) and a sense of school connectedness and belonging (i.e., school-level) may contribute to resilience in adolescents. At the neighborhood level, community-based intervention programs that support the creation of strong social cohesion and neighborhood connections may be important to foster youth resilience. For these multi-level interventions to be implemented successfully, cross-sector and inter-group collaboration (e.g., families, schools, child welfare agencies, community organizations) will be critical. Similarly, policymakers should acknowledge the importance of a socioecological systems framework and allocate increased funding and resources to support programs and services that target multi-level environmental contexts to promote positive and resilient development of young people who have experienced childhood maltreatment.

## 8. Conclusions

The current study has shown that positive environmental contexts can serve as important protective factors that promote resilient development among adolescents with a history of childhood trauma. Positive maternal involvement appears to make salient contributions to adolescents’ inner well-being characterized by positive psychological characteristics (engagement, perseverance, optimism, connectedness, happiness). In addition to the family context (i.e., positive involvement of the mother), school and neighborhood contexts such as school connectedness and neighborhood social cohesion may play important roles in the development of adolescent social skills. Collectively, the findings suggest that environmental protective factors could help adolescents overcome childhood adversity and strengthen their intra- and inter-personal capabilities and competencies after exposure to maltreatment in childhood.

## Figures and Tables

**Table 1 children-10-00001-t001:** Sample characteristics and descriptive statistics of study variables (N = 1729).

	%	M (SD)	Range
Positive adolescent functioning at age 15		68.53 (6.37)	36–80
Social skills at age 15		17.01 (3.85)	3–24
Father’s involvement at age 9		1.86 (0.72)	0–3
Mother’s involvement at age 9		2.19 (0.51)	0–3
School connectedness at age 9		3.08 (0.97)	0–4
Social cohesion at age 9		2.97 (0.69)	1–4
Informal social control at age 9		3.24 (0.82)	1–4
Child gender (male)	51.53		
Child race/ethnicity			
White	18.74		
Black	46.44		
Hispanic	18.68		
Multiracial	14.81		
Other Race	1.33		
Mother’s age		25.30 (5.96)	15–43
Mother’s education level (less than high school)	19.26		
Mother’s marital status	43.03		
Mother’s # of children		2.70 (1.31)	0–8
Economic hardship (low SES)		1.33 (1.66)	0–9

**Table 2 children-10-00001-t002:** Correlations among key study variables.

	1	2	3	4	5	6	7
1. Positive adolescent functioning	---						
2. Adolescent social skills	0.48 ***	---					
3. Father’s involvement	0.04	0.08 ***	---				
4. Mother’s involvement	0.08 ***	0.08 ***	0.25 ***	---			
5. School connectedness	0.05 *	0.10 ***	0.13 ***	0.21 ***	---		
6. Social cohesion	0.05 *	0.12 ***	0.08 **	0.02	0.05 *	---	
7. Informal social control	0.03	0.09 ***	0.03	0.03	0.01	0.52 ***	---

Note. * *p* < 0.05. ** *p* < 0.01. *** *p* < 0.001.

**Table 3 children-10-00001-t003:** Predictors of positive adolescent functioning and adolescent social skills (N = 1729).

	Positive Adolescent Functioning			Adolescent Social Skills		
	B	SE	*p*	B	SE	*p*
Father’s involvement	0.22	0.22	0.324	0.23	0.13	0.085
Mother’s involvement	**0.86**	**0.32**	**0.006**	**0.37**	**0.19**	**0.049**
School connectedness	0.28	0.16	0.088	**0.31**	**0.10**	**0.001**
Social cohesion	0.37	0.26	0.161	**0.38**	**0.16**	**0.015**
Informal social control	0.12	0.22	0.570	0.11	0.13	0.382
Child gender (male)	**1.46**	**0.31**	**0.000**	0.27	0.18	0.145
Child race/ethnicity						
Black	**1.56**	**0.45**	**0.001**	**−0.55**	**0.27**	**0.040**
Hispanic	0.29	0.52	0.580	**−0.75**	**0.31**	**0.017**
Multiracial	0.86	0.54	0.112	0.22	0.33	0.491
Other	−1.24	1.36	0.365	−0.13	0.82	0.875
Mother’s age	−0.03	0.03	0.315	−0.01	0.02	0.541
Mother’s education level (less than high school)	−0.11	0.40	0.790	**−0.55**	**0.24**	**0.024**
Mother’s marital status (married)	−0.34	0.34	0.321	0.13	0.20	0.535
Mother’s # of children	−0.11	0.12	0.348	**−0.20**	**0.07**	**0.004**
Economic hardship (low SES)	−0.13	0.10	0.176	**−0.13**	**0.06**	**0.028**

Note. For child’s race/ethnicity, White was the reference group. Bolded numbers indicate statistically significant findings.

## Data Availability

The data are publicly available through Fragile Families. https://fragilefamilies.princeton.edu, accessed on 12 October 2022.

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
