# Peer review of "Family-, School-, and Neighborhood-Level Predictors of Resilience for Adolescents with a History of Maltreatment"

_children, 2022, doi:10.3390/children10010001_

Round 1

Reviewer 1 Report

I think it is an excellent cohort study with numerous results. However, I feel that the submitted manuscript lacks originality and importance. I would suggest that you could improve your methods and devise your writing. Because you have achieved so much, you must distinguish your content from published articles; I recommend using IThenticate.

Foreword.

This chapter was long, and it was like reading a review article. As the author himself acknowledges, there is a lot of previous research. Please clearly indicate how you set up the hypotheses for this study in light of the previous findings.

Regarding the definition of resilience, it was unclear why it consists of positive adolescent social functioning and social skills. Please explain the relationship between the concepts.

Methods.

You are currently doing a multiple regression analysis; have you considered a multilevel analysis?

Results/discussion

I suppose ample studies have pointed out the mother's involvement. The discussion could be a little more creative.

Regarding being Black, there seem to be conflicting results for positive adolescent social functioning and social skills. How do you interpret this?

There is a lack of consideration of school- and neighbourhood-level findings.

Author Response

Thank you so much for your valuable feedback.

Reviewer 2 Report

The study contributes to the understanding of resilience predictors in adolescents with a history of maltreatment. It is a relevant study because it addresses a global public health problem with consistency, requiring minor adjustments.

ABSTRACT

It is well structured and written in a concise and easy-to-read form, highlighting the main points of the article.

INTRODUCTION

It presents the study problem in a concise and well-structured manner, placing the significance of the study based on relevant and updated literature. The theoretical basis of the study is described in a very consistent way. Defines the objectives of the study adequately.

METHODS

They were described in a detailed and clear way.

Page 7 - It is only necessary to describe what the acronym PSID-CDS III means, which is only described in the references.

RESULTS

The results are well organized and well presented in tables.

Page 8 – Lines 392 a 395: There are 2 incoherent sentences that need to be clarified: “The majority of the mothers in the sample (80.7%) had a higher level of education than high school...”. “About 84.2% of the mothers of the focal child have attained a higher level of education than high school education,...” What is the correct sentence? According to the table, it would be the first

Page 8 – Line 396: the percentage that is in the following sentence “...38% of mothers were married when the focal child was age 9.” it is not what the table indicates.

DISCUSSION

It is well structured, dialogues the results with the previously and updated published literature. The authors discuss the advantages, limitations, and implications of the study adequately.

REFERENCES

References are up to date and well organized.

Author Response

(The authors gave the same response as above.)

Round 2

Reviewer 1 Report

Thank you very much for your sincere correction. The response letter provides a logical explanation. The corrections are good. I believe that your supplementary explanation has increased the significance of your paper and its contribution to the scientific community.